

# A noniteratiave approach to modelling moist thermodynamics

Nadya Moisseeva[1] and Roland Stull[1]

[1]Dept. of Earth, Ocean and Atmospheric Sciences,University of British Columbia, 2020-2207 Main Mall Vancouver, BC, Canada V6T 1Z4

*Correspondence to:* Nadya Moisseeva (nmoisseeva@eoas.ubc.ca)

**Abstract.** Formulation of noniterative mathematical expressions for moist thermodynamics presents a challenge for both numerical and theoretical modellers. This technical note offers a simple and efficient tool for approximating two common thermodynamic relationships: temperature $T$ at a given pressure $P$ along a saturated adiabat $T(P, \theta_w)$, as well as its corresponding inverse form $\theta_w(P, T)$, where $\theta_w$ is wet-bulb potential temperature. Our method allows direct calculation of $T(P, \theta_w)$ and $\theta_w(P, T)$ on a thermodynamic domain bounded by $-70 \leq \theta_w < 40$°C, $P > 1$ kPa and $-100 \leq T < 40$°C, $P > 1$ kPa, respectively. The proposed parameterizations offer high accuracy (mean absolute errors of 0.017°C and 0.002°C for $T(P, \theta_w)$ and $\theta_w(P, T)$, respectively) on a notably larger thermodynamic region than previously studied. The paper includes a method summary, as well as a ready-to-use tool to aid atmospheric physicists in their practical applications.

## 1 Introduction

Saturated thermodynamics commonly present a challenge for theoretical studies because moist convective condensation, such as deep cumulus precipitation, often involves pseudoadiabtic (irreversible) processes. The latent heat released during water vapour condensation is important for estimating thunderstorm intensity and thickness, precipitation amount and phase, global climate and atmospheric general circulation (Stull, 2017). These processes are governed by nonlinear equations that require iteration to solve. Numerical weather prediction (NWP) models, hence, suffer from the added computational cost to their cloud, precipitation, convection and turbulence schemes and parameterizations, because of the iterations required during each timestep of the NWP integration.

A common iterative approach, such as described by Caballero (2014), uses step-wise numerical integration along a saturated adiabat for any constant wet-bulb potential temperature $\theta_w$. The moist adiabatic lapse rate is derived from conservation of moist entropy as a function of temperature $T$ and saturated mixing ratio $r_s$, which itself is a nonlinear function of $T$ and pressure $P$. To imporove efficiency Davies-Jones (2008) proposed a different iterative method, based on inverting Bolton's formula for equivalent potential temperature valid for the pressure range $10 \leq P \leq 105$ kPa and wet-bulb potential temperatures $-20 < \theta_w < 40$°C. As a valuable noniterative alternative, Bakhshaii and Stull (2013) offer an approximate solution devised using gene-expression programming (GEP). They provide two separate sets of equations for determining $T(P, \theta_w)$ and $\theta_w(P, T)$, for the domain bounded by $-30 < \theta_w < 40$°C, $P > 20$ kPa and $-60 < T < 40$°C. The complex nature of the problem required their splitting of the modelled region into sub-domains, resulting in error discontinuity. The method also produced fairly large





errors (on the order of a degree) in the upper atmosphere. Despite the limitations, to our knowledge Bakhshaii and Stull (2013) is the only existing noniterative solution to approximate saturated pseudoadiabats.

Our current study presents a different approach for directly calculating $T(P, \theta_w)$ and $\theta_w(P, T)$ offering improved accuracy for a larger thermodynamic domain. The method, described in Section 2, normalizes the raw data before fitting it with polyno-

mials. The resultant approximation is evaluated against the "truth" (the iterated solution) and summarized in Sections 3 and 4, respectively. As Supplementary Material we offer the readers a ready-to-use spreadsheet implementing our methodology.

The goal of this paper is to provide a simple tool that can aid analytical modellers in their theoretical work as well as numerical modellers in reducing the computational cost of their simulations.

## 2   Method Description

**2.1   Data**

In order to obtain a set of "truth" curves for $T(P, \theta_w)$ we have used an iterative approach to numerically integrate the equation for $\frac{dT}{dP}$ (Tables 1 and 2) for values in the range of $-100 \leq \theta_w < 100$°C between $105 \geq P > 1$ kPa. We found that numerical integration along a saturated adiabat $\theta_w$ from the bottom to the top of the domain required an increasingly refined pressure step, as all adiabats tend to absolute zero near the top of the atmosphere, and each consecutive pressure step corresponds to a larger

temperature jump. For our numerical integration we used $10^{-4}$ kPa step for $105 \geq P > 10$ kPa, $10^{-5}$ kPa step for $10 \geq P > 2$ kPa and $10^{-6}$ kPa step for $2 \geq P > 1$ kPa. The resulting curves (shown on thermo diagram in Figure 1) are taken as "truth", to which we fit our polynomial-based optimization. The non-iterative approximations for $T(P, \theta_w)$ and $\theta_w(T, \theta_w)$ described below are valid for thermodynamic ranges bounded by $-70 \leq \theta_w < 40$°C and $-100 \leq T < 40$°C, respectively.

### 2.2   Approximating $T(P, \theta_w)$

While the moist adiabiatic curves $\theta_w$ in Figure 1 look smooth and fairly similar, it is challenging for most common optimization routines to capture all of them with a single analytical expression. To remove some of the inherent nonlinearity in the data we can normalize our curves by dividing each $\theta_w$ by a reference moist aidiabat $\theta_{ref}$. For our example we used $\theta_{ref} = -70$°C. This particular choice of $\theta_{ref}$ implies no theoretical importance. It is possible to choose any of the directly calculated normalized adiabats to represent $\theta_{ref}$. Depending on the choice, the resulting transformed adiabats shift around the $\theta_{ref}$ unity line. The

single consideration for choosing a particular $\theta_{ref}$ is the ease and accuracy with which it can be fit by a particular optimization tool.

We use polynomial fitting to describe $T(P)$ for the fixed $\theta_{ref}$. This is convenient, since polynomials are generally well-behaved. The choice of the degree of polynomial depends on the desired precision level. Since we are examining a fixed range of temperatures relevant to atmospheric applications, the potentially chaotic behavior of high-degree polynomials outside of

the modelled domain is not a primary concern. For this example, the aim was to ensure that the mean absolute error (MAE) is





on the order of $10^{-2}$ degrees C, requiring a 20th degree polynomial to achieve such fit. The true and modelled $\theta_{ref} = -70°C$ can be seen in Fig. 2 with fit coefficients provided in Table 3.

The next step is to choose a single functional form to represent the entire family of the transformed curves. Each given shape of a particular curve is then controlled by variable parameters of the same function. A number of simple functions

exists that are able to model the above relationship. For this work we tested bi-exponential, arctan, rational and polynomial functions. Generally, a reasonable fit can be achieved with both bi-exponential and arctan functions using as little as three variable parameters. While efficient, the results of such fit are unlikely to be sufficiently accurate to be useful for real-life modelling applications. Another concern with these choices is that the variable parameters are not well-behaved functions and are hence difficult to model.

Polynomial fitting doesn't appear to suffer from such issues. Moreover, the accuracy can be controlled by changing the degree of the polynomial and, hence, allowing a higher number of variable parameters. In this example, the curves were modelled using 10th degree polynomials, resulting in 11 variable parameters. Conveniently, and unlike other functional forms mentioned above, these parameters are also well-behaved. They can, again, be modelled using high-degree polynomials to the desired level of accuracy. Results of parameter fitting for this given example were again produced using 20th degree

polynomials and can be see in Figure 3 with fit coefficients provided in Table 4. The resulting modelled moist adiabats can be seen in Figure 1, compared to the truth values.

## 2.3    Approximating $\theta_w(P,T)$

A similar approach can be used to produce a non-iterative approximation for $\theta_w(P,T)$. To obtain a new set of curves representing lines of constant temperature in $\theta_w$ domain, we have used our existing dataset for $-100 \leq \theta_w < 100°C$ to extract isotherms

on a 0.5°C and 0.1 kPa grid for $-100 \leq T < 40°C$ and $105 \geq P > 1$ kPa.

Similarly to our earlier approach, we select a single reference curve $T_{ref} = T_{-100°C}$ and use a high-order polynomial to model it as a function of pressure (Figure 4, Table 5). We then produce a set of transformed curves by normalizing the isotherms with $T_{ref}$. We fit the transformed curves with 10th degree polynomials, obtaining a dataset for 11 variable parameters. Finally, we use polynomials to model the variable parameters (Figure 5, Table 6). The following section discusses the results and

accuracy of our optimization procedure.

## 3    Evaluation

To test the accuracy of the proposed method, we compared our modelled curves for $T(P,\theta_w)$ and $\theta_w(P,T)$ with those obtained through direct calculation (the "truth" iterative solution). The results of the evaluation for $T(P,\theta_w)$ are shown in Figure 6, indicating errors on the order of few hundredths of a degree throughout most of the domain. Warmer values near the top of

the domain tend to be modelled least accurately. Mean absolute error (MAE) for the entire modelled thermodynamic region is 0.017°C. Error contours for $\theta_w(P,T)$ are shown in Figure 7, with errors on the order of few thousandths of a degree throughout most of the domain and overall MAE = 0.002°C. Once again, values near the low-pressure limit tend to be least





accurate. Notably, applying the above optimization on a slightly shallower pressure domain of $P > 2$ kPa, allows to improve the overall MAE for both approximations by an additional order of magnitude.

As mentioned earlier, improved accuracy may also be achieved with the use of even higher degrees of polynomials for parameter fits. However, such precision is unlikely to be necessary, as some of the thermodynamic relationships used in the "truth" iterative computations contain substantially larger errors, than those introduced by the above optimization procedure (Davies-Jones, 2009; Koutsoyiannis, 2012). Moreover, conventional pseudoadiabatic diagrams, such as those used by U.S. Air Force (USAF), Environment Canada (EC) and Air Transport Association of America (ATAA), differ by nearly 1°C at the 20 kPa pressure level (Bakhshaii and Stull, 2013).

Though the upper 10 kPa of the atmosphere contains the largest errors with our proposed approach, this vertical subrange also presents the most significant challenge for direct (iterative) numerical modelling. Accurate numerical computation requires an increasingly refined vertical step for the top part of the atmosphere. Hence, despite the errors, the proposed approximation offers a more accurate solution than one would obtain with direct iterative approach using a somewhat coarse yet computationally demanding 0.001 kPa pressure step.

While common weather phenomena generally remain in the troposphere, the validity of the current method on a notably larger vertical domain is particularly useful in the lower latitudes. Deep vertical extent of tropical thunderstorms, hurricanes and typhoons in combination with the high tropopause altitude in the tropics (10-15 kPa) contribute to large computational costs of modelling these potentially destructive events.

## 4   Summary of approach

Individual steps to directly compute $T(P, \theta_w)$ and $\theta_w(P, T)$ are summarized below. This sample procedure, along with the required coefficient tables are provided in a ready-to-use form in the attached spreadsheet (Supplementary Material). Note, that the same coefficients presented in Tables 3 - 6 are rounded to fewer significant digits to fit them and might, hence, offer lower accuracy, relative to the full significant digits in the suplementary spreadsheet.

### 4.1   Computing $T(P, \theta_w)$

Let $n = 0, ..., 10$ correspond to the index of individual polynomial coefficients and $m = 20$ be the degree of polynomial fits for $\theta_{ref}(P)$ and $k_n(\theta_w)$, respectively.

1) Compute coefficients $k_n(\theta_w)$ using polynomial coefficients $a_{20}, ..., a_0$ in Table I in Supplementary Material (and Table 4 here) :

$$k_n(\theta_w) = \sum_{i=0}^{m} a_{(n, m-i)} \theta_w^{m-i} \tag{1}$$

for $\theta_w$ in degrees C.





2) Compute $\theta_{ref}(P)$ using polynomial coefficients $b_{20}, ..., b_0$ in Table II in Supplementary Material (and Table 3 here):

$$\theta_{ref}(P) = \sum_{j=0}^{m} b_{(m-j)} P^{m-j} \tag{2}$$

for $P$ in kPa.

3) Compute $T(\theta_{ref})$:

$$T(P, \theta_w) = T(\theta_{ref}) = \sum_{h=0}^{n} k_h \theta_{ref}^{n-h} \tag{3}$$

where $T$ and $\theta_{ref}$ are in Kelvins, and values of $k_{0,...,n}$ correspond to polynomial coefficients calculated in Step 1.

## 4.2 Computing $\theta_w(P, T)$

Let $n = 0, ..., 10$ correspond to the index of individual polynomial coefficients and $m = 20$ be the degree of polynomial fits for $T_{ref}(P)$ and $\kappa_n(T)$, respectively.

1) Compute coefficients $\kappa_n(T)$ using polynomial coefficients $\alpha_{20}, ..., \alpha_0$ in Table III in Supplementary Material (Table 6):

$$\kappa_n(T) = \sum_{i=0}^{m} \alpha_{(n, m-i)} T^{m-i} \tag{4}$$

for $T$ in degrees C.

2) Compute $T_{ref}(P)$ using polynomial coefficients $\beta_{20}, ..., \beta_0$ in Table IV in Supplementary Material (Table 5):

$$T_{ref}(P) = \sum_{j=0}^{m} \beta_{(m-j)} P^{m-j} \tag{5}$$

for $P$ in kPa.

3) Compute $\theta_w(T_{ref})$:

$$\theta_w(P, T) = \theta_w(T_{ref}) = \sum_{h=0}^{n} \kappa_h T_{ref}^{n-h} \tag{6}$$

where $\theta_w$ and $T_{ref}$ are in degrees C, and values of $\kappa_{0,...,n}$ correspond to polynomial coefficients calculated in Step 1.

## 5 Usage Example

Meteorologists typically use both $\theta_w(P, T)$ and $T(P, \theta_w)$ for moist convection such as thunderstorms, frontal clouds, mountain-wave clouds, and many other phenomena where a saturated air parcel moves vertically. Cloud base of convective clouds marks the bottom of saturated ascent, and cloud top marks the top.

For example, suppose that the forecast at some tropical weather station is $P = 100$ kPa, $T = 32°C$ with dewpoint $T_d = 21°C$ (corresponding to a water vapor mixing ratio of approximately $r = 16$ g kg$^{-1}$). Further suppose that a force (e.g., buoyancy,





frontal uplift, or orographic uplift) causes an air parcel with these initial conditions to rise. Initially this air parcel is unsaturated (not cloudy), so we don't need to use the polynomial or iterative equations. Instead, simpler non-iterative equations apply for the thermodynamic state as the parcel rises dry adiabatically. Namely, its temperature cools at the dry adiabatic lapse rate ($9.8°C\ km^{-1}$), and the mixing ratio and potential temperature are constant. This air parcel will become saturated (i.e., cloud

5   base) at the lifting condensation level (LCL). With this information, other thermodynamic equations (Stull, 2017) can be used to find conditions at the LCL: $z_{LCL} = 1.375$ km, $P_{LCL} = 85.4$ kPa, and $T_{LCL} = 18.5°C$.

Given this initial $P$ and $T$ at the LCL, we can use the polynomial equations provided in this paper to compute which moist adiabat the cloudy air parcel will follow: $\theta_w(P, T) = 24.0°C$.

If this cloudy air parcel (still following the $\theta_w(P, T) = 24.0°C$ adiabat) rises to an altitude where the pressure is $P = 24.0$

10   kPa, then we can use the second set of polynomial equations in this paper to find the final temperature of the air parcel at this new height: $T(P, \theta_w) = -39.8°C$.

## 6   Discussion and Conclusions

The polynomial method proposed here is accurate, smooth, and computationally efficient. For example, given the cloud base and cloud top pressures of the previous example, the tally of computer operations to find both the initial and the final tempera-

ture are: 230 additions and subtractions, 2365 multiplies (where rational numbers to integer powers are counted as sequential multiplies). Compare that to the computation tally for the "truth" iterative solution, requiring a total of 2,750,000 variable pressure steps, where each step has: 8 additions and subtractions, 17 multiplies (where rational numbers to integer powers are counted as sequential multiplies), 9 divides, and 2 math functions (e.g., log, exp, non-integer exponents), totalling to 988,200,000 operations from the bottom to the top of the domain.

Also, for comparison, some numerical weather prediction models use a look-up table to get the average saturated adiabatic lapse rate $\Delta\theta_w/\Delta P$ as a function of $P$ and $T$. While this method is fairly fast, it is also less accurate, and approximates the saturated lapse rate as a series of short straight-line segments instead of a smooth curve. It also has discontinuous jumps of saturated lapse rate as $T$ varies along an isobar.

Thus, the polynomial method proposed here provides a computation of high accuracy and smooth variation across the

whole thermodynamic diagram range, at intermediate computation speed compared to the other methods. Moreover, it helps to model moist thermodynamics on a wider temperature range with roughly two orders of magnitude MAE improvement over the existing solution.

In addition to the reduced computational costs of obtaining solutions for $T(P, \theta_w)$ and $\theta_w(P, T)$ in numerical simulations and improving accuracy, we hope that our tool will aid analytical modellers in their theoretical work.

## 30   7   Tables



**Table 1.** Table of constants

| Constant | Description [units] |
| --- | --- |
| $R_d = 287.058$ | gas constant for dry air [J K$^{-1}$ kg$^{-1}$] |
| $R_v = 461.5$ | gas constant for water vapour [J K$^{-1}$ kg$^{-1}$] |
| $C_{pd} = 1006$ | specific heat of dry air at constant pressure [J K$^{-1}$ kg$^{-1}$] |
| $T_0 = 273.16$ | reference temperature [K] |
| $P_0 = 100.$ | reference pressure [kPa] |
| $e_0 = 0.611657$ | Clausius-Clayperon constant [kPa] |
| $\varepsilon = \frac{R_d}{R_v} = 0.6220$ | ratio of gas constants [kg kg$^{-1}$] |





**Table 2.** Variable definitions

| Variable | Description [units] |
| --- | --- |
| $T$ | [K] ambient temperature |
| $P$ | [kPa] pressure |
| $\theta_w$ | [K] saturated adiabat where the value of $T$ defined at $P = P_0$ is defined as wet-bulb potential temperature |
| $e_s = e_0 exp\left[24.921\left(1 - \frac{T_0}{T}\right)\right]\left(\frac{T_0}{T}\right)^{5.06}$ | [kPa] saturation vapour pressure |
| $L_v = 3.139 \times 10^6 - 2336 * T$ | [K] latent heat of vapourization (Koutsoyiannis, 2012) |
| $r_s = \varepsilon\frac{e_s}{(P - e_s)}$ | [kg kg$^{-1}$] saturation mixing ratio |
| $\frac{dT}{dP} = \frac{\frac{R_d}{C_{pd}}T + \frac{L_v}{C_{pd}}r_s}{P(1 + \frac{\frac{L_v^2}{R_v C_{pd}}r_s}{T^2})}$ | [K kPa$^{-1}$] change of temperature with pressure along a saturated adiabat, which can be iterated to find $T$ vs. $P$ |





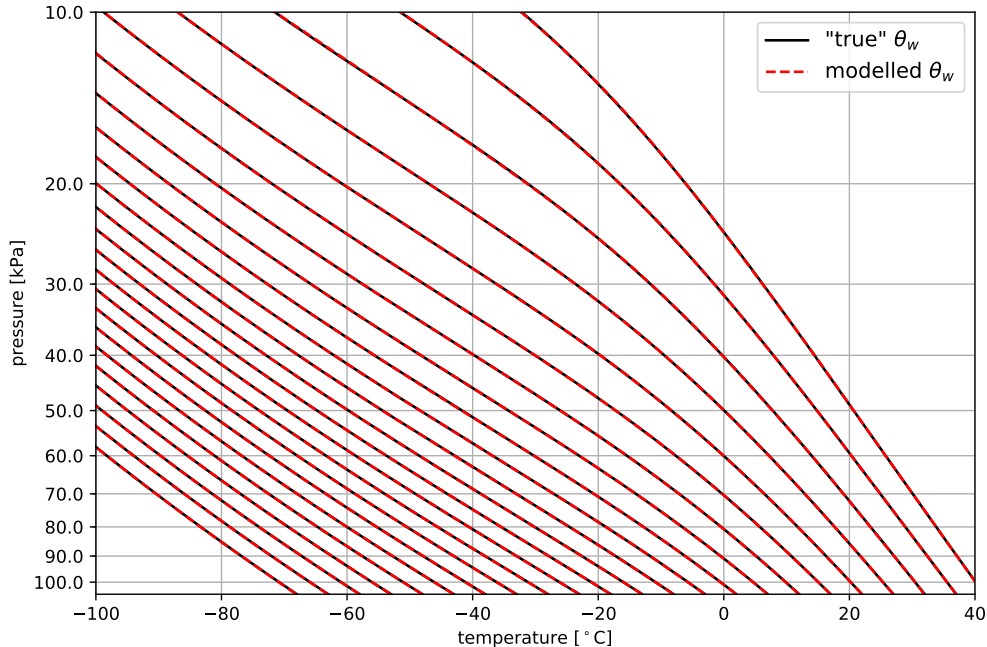

**Figure 1.** Emagram plot showing select "true" (solid black) and modelled (dashed red) moist adiabats $\theta_w$ (difference not apparent at this scale). Temperature and pressure domains are restricted for clarity. An emagram (energy mass diagram) is a thermodynamic diagram with the log of pressure on the vertical axis, plotted with max and min values reversed, so that higher in the diagram corresponds to higher in the atmosphere, where pressures are lower. The non-iterative results presented in this paper can be plotted on any thermodynamic diagram, including tephigrams and skew-T diagrams.

## 8   Figures





**Table 3.** Polynomial coefficients for fitting $\theta_{ref}(P)$.

| $b_{20}$ | $b_{19}$ | $b_{18}$ | $b_{17}$ | $b_{16}$ | $b_{15}$ | $b_{14}$ | $b_{13}$ | $b_{12}$ | $b_{11}$ | $b_{10}$ | $b_9$ | $b_8$ | $b_7$ | $b_6$ | $b_5$ | $b_4$ | $b_3$ | $b_2$ | $b_1$ | $b_0$ |
|---|---|---|---|---|---|---|---|---|---|---|---|---|---|---|---|---|---|---|---|---|
| -1.83e-32 | 1.47e-29 | -5.06e-27 | 9.24e-25 | -8.30e-23 | -2.61e-22 | 8.87e-19 | -7.61e-17 | -2.57e-15 | 1.16e-12 | -1.29e-10 | 8.69e-09 | -4.06e-07 | 1.37e-05 | -3.35e-04 | 5.95e-03 | -7.55e-02 | 6.71e-01 | -4.12e+00 | 2.00e+01 | 3.84e+01 |

**Table 4.** Polynomial coefficients for fitting parameters $k_0 \ldots k_{10}(\theta_w)$.

| Coefficient | $\alpha_0$ | $\alpha_1$ | $\alpha_2$ | $\alpha_3$ | $\alpha_4$ | $\alpha_5$ | $\alpha_6$ | $\alpha_7$ | $\alpha_8$ | $\alpha_9$ | $\alpha_{10}$ | $\alpha_{11}$ | $\alpha_{12}$ | $\alpha_{13}$ | $\alpha_{14}$ | $\alpha_{15}$ | $\alpha_{16}$ | $\alpha_{17}$ | $\alpha_{18}$ | $\alpha_{19}$ | $\alpha_{20}$ |
|---|---|---|---|---|---|---|---|---|---|---|---|---|---|---|---|---|---|---|---|---|---|
| $k_0$ | -3.06e-18 | -1.04e-19 | 9.97e-22 | 1.24e-21 | 4.95e-23 | 1.30e-24 | -7.78e-26 | -1.39e-26 | -4.66e-28 | 1.33e-29 | 1.05e-30 | 8.59e-33 | -7.07e-34 | -1.74e-35 | 7.96e-38 | 7.65e-39 | 7.84e-41 | -6.69e-43 | -2.03e-44 | -1.62e-46 | -4.57e-49 |
| $k_1$ | 4.11e-15 | 1.26e-16 | -1.39e-17 | -1.63e-18 | -6.24e-20 | -1.55e-21 | 1.08e-22 | 1.83e-23 | 5.97e-25 | -1.82e-26 | -1.39e-27 | -1.07e-29 | 9.45e-31 | 2.28e-32 | -1.12e-34 | -1.02e-35 | -1.03e-37 | 9.07e-40 | 2.70e-41 | 2.14e-43 | 6.04e-46 |
| $k_2$ | -2.45e-12 | -6.68e-14 | 8.48e-15 | 9.41e-16 | 3.46e-17 | 8.08e-19 | -6.63e-20 | -1.06e-20 | -3.38e-22 | 1.09e-23 | 8.05e-25 | 5.82e-27 | -5.58e-28 | -1.32e-29 | 7.02e-32 | 5.98e-33 | 5.94e-35 | -5.44e-37 | -1.59e-38 | -1.26e-40 | -3.54e-43 |
| $k_3$ | 8.50e-10 | 2.06e-11 | -2.99e-12 | -3.15e-13 | -1.11e-14 | -2.43e-16 | 2.35e-17 | 3.59e-18 | 1.11e-19 | -3.83e-21 | -2.72e-22 | -1.83e-24 | 1.92e-25 | 4.44e-27 | -2.54e-29 | -2.04e-30 | -2.00e-32 | 1.90e-34 | 5.45e-36 | 4.28e-38 | 1.20e-40 |
| $k_4$ | -1.90e-07 | -4.06e-09 | 6.73e-10 | 6.79e-11 | 2.31e-12 | 4.65e-14 | -5.37e-15 | -7.81e-16 | -2.35e-17 | 8.63e-19 | 5.92e-20 | 3.68e-22 | -4.24e-23 | -9.62e-25 | 5.91e-27 | 4.49e-28 | 4.33e-30 | -4.26e-32 | -1.20e-33 | -9.40e-36 | -2.64e-38 |
| $k_5$ | 2.87e-05 | 5.37e-07 | -1.01e-07 | -9.79e-09 | -3.21e-10 | -5.91e-12 | 8.20e-13 | 1.14e-13 | 3.32e-15 | -1.31e-16 | -8.65e-18 | -4.92e-20 | 6.29e-21 | 1.40e-22 | -9.22e-25 | -6.64e-26 | -6.29e-28 | 6.43e-30 | 1.78e-31 | 1.39e-33 | 3.88e-36 |
| $k_6$ | -2.95e-03 | -4.83e-05 | 1.03e-05 | 9.60e-07 | 3.03e-08 | 5.05e-10 | -8.49e-11 | -1.13e-11 | -3.19e-13 | 1.34e-14 | 8.58e-16 | 4.41e-18 | -6.35e-19 | -1.38e-20 | 9.75e-23 | 6.67e-24 | 6.20e-26 | -6.59e-28 | -1.79e-29 | -1.39e-31 | -3.87e-34 |
| $k_7$ | 2.04e-01 | 2.92e-03 | -7.01e-04 | -6.31e-05 | -1.92e-06 | -2.85e-08 | 5.89e-09 | 7.54e-10 | 2.06e-11 | -9.23e-13 | -5.70e-14 | -2.61e-16 | 4.30e-17 | 9.13e-19 | -6.90e-21 | -4.49e-22 | -4.10e-24 | 4.52e-26 | 1.21e-27 | 9.32e-30 | 2.59e-32 |
| $k_8$ | -9.11e+00 | -1.14e-01 | 3.04e-02 | 2.66e-03 | 7.85e-05 | 1.01e-06 | -2.62e-07 | -3.22e-08 | -8.47e-10 | 4.07e-11 | 2.43e-12 | 9.73e-15 | -1.86e-15 | -3.87e-17 | 3.12e-19 | 1.93e-20 | 1.73e-22 | -1.99e-24 | -5.21e-26 | -4.01e-28 | -1.11e-30 |
| $k_9$ | 2.37e+02 | 2.59e+00 | -7.63e-01 | -6.49e-02 | -1.86e-03 | -2.04e-05 | 6.71e-06 | 7.97e-07 | 2.02e-08 | -1.04e-09 | -6.00e-11 | -2.06e-13 | 4.67e-14 | 9.50e-16 | -8.14e-18 | -4.83e-19 | -4.25e-21 | 5.07e-23 | 1.30e-24 | 9.99e-27 | 2.77e-29 |
| $k_{10}$ | -2.69e+03 | -2.59e+01 | 8.38e+00 | 6.96e-01 | 1.94e-02 | 1.74e-04 | -7.56e-05 | -8.67e-06 | -2.11e-07 | 1.16e-08 | 6.50e-10 | 1.85e-12 | -5.14e-13 | -1.02e-14 | 9.30e-17 | 5.29e-18 | 4.57e-20 | -5.66e-22 | -1.43e-23 | -1.09e-25 | -3.02e-28 |

**Table 5.** Polynomial coefficients for fitting $T_{ref}(P)$.

| $\beta_{20}$ | $\beta_{19}$ | $\beta_{18}$ | $\beta_{17}$ | $\beta_{16}$ | $\beta_{15}$ | $\beta_{14}$ | $\beta_{13}$ | $\beta_{12}$ | $\beta_{11}$ | $\beta_{10}$ | $\beta_9$ | $\beta_8$ | $\beta_7$ | $\beta_6$ | $\beta_5$ | $\beta_4$ | $\beta_3$ | $\beta_2$ | $\beta_1$ | $\beta_0$ |
|---|---|---|---|---|---|---|---|---|---|---|---|---|---|---|---|---|---|---|---|---|
| 1.23e-33 | -1.07e-30 | 4.02e-28 | -8.18e-26 | 8.71e-24 | -1.64e-22 | -8.05e-20 | 9.54e-18 | 9.37e-16 | -1.18e-13 | 1.61e-11 | -1.25e-09 | 6.61e-08 | -2.49e-06 | 6.74e-05 | -1.31e-03 | 1.81e-02 | -1.76e-01 | 1.18e+00 | -7.51e+00 | 5.64e+01 |

**Table 6.** Polynomial coefficients for fitting parameters $k_0 \ldots k_{10}(T)$.

| Coefficient | $\alpha_0$ | $\alpha_1$ | $\alpha_2$ | $\alpha_3$ | $\alpha_4$ | $\alpha_5$ | $\alpha_6$ | $\alpha_7$ | $\alpha_8$ | $\alpha_9$ | $\alpha_{10}$ | $\alpha_{11}$ | $\alpha_{12}$ | $\alpha_{13}$ | $\alpha_{14}$ | $\alpha_{15}$ | $\alpha_{16}$ | $\alpha_{17}$ | $\alpha_{18}$ | $\alpha_{19}$ | $\alpha_{20}$ |
|---|---|---|---|---|---|---|---|---|---|---|---|---|---|---|---|---|---|---|---|---|---|
| $k_0$ | 1.23e-18 | -9.37e-21 | -7.59e-22 | 1.48e-22 | 4.32e-24 | -3.27e-25 | -2.72e-26 | -2.23e-29 | 4.47e-29 | 8.90e-31 | -2.42e-32 | -1.02e-33 | -3.72e-36 | 3.76e-37 | 7.09e-39 | 8.95e-42 | -1.32e-42 | -2.09e-44 | -1.55e-46 | -5.91e-49 | -9.39e-52 |
| $k_1$ | 4.30e-16 | 2.14e-19 | -3.30e-19 | 4.47e-20 | 1.70e-21 | -1.07e-22 | -9.21e-24 | -1.14e-26 | 1.48e-26 | 3.02e-28 | -7.88e-30 | -3.43e-31 | -1.35e-33 | 1.25e-34 | 2.39e-36 | 3.43e-39 | -4.38e-40 | -7.00e-42 | -5.21e-44 | -1.99e-46 | -3.17e-49 |
| $k_2$ | 5.74e-14 | 5.50e-16 | -4.65e-17 | 4.30e-18 | 2.37e-19 | -1.16e-20 | -1.09e-21 | -2.50e-24 | 1.69e-24 | 3.61e-26 | -8.70e-28 | -3.99e-29 | -1.81e-31 | 1.42e-32 | 2.81e-34 | 4.97e-37 | -5.02e-38 | -8.16e-40 | -6.11e-42 | -2.35e-44 | -3.75e-47 |
| $k_3$ | 3.55e-12 | 8.40e-14 | -2.46e-15 | 6.10e-17 | 1.20e-17 | -2.86e-19 | -4.04e-20 | -2.75e-22 | 5.63e-23 | 1.43e-24 | -2.49e-26 | -1.43e-27 | -9.62e-30 | 4.67e-31 | 1.04e-32 | 3.08e-35 | -1.69e-36 | -2.92e-38 | -2.24e-40 | -8.78e-43 | -1.42e-45 |
| $k_4$ | 8.40e-14 | 6.41e-12 | 3.11e-14 | -1.32e-14 | -1.45e-15 | 2.95e-17 | 1.58e-18 | -1.36e-20 | -2.92e-21 | -4.23e-23 | 1.83e-24 | 5.99e-26 | 1.42e-29 | -2.49e-29 | -3.98e-31 | 3.36e-34 | 8.57e-35 | 1.24e-36 | 8.79e-39 | 3.25e-41 | 5.04e-44 |
| $k_5$ | 9.05e-11 | 3.60e-10 | 5.50e-13 | -6.55e-13 | -3.08e-14 | 1.71e-15 | 1.40e-16 | -3.43e-20 | -2.19e-19 | -4.33e-21 | 1.17e-22 | 5.02e-24 | 1.94e-26 | -1.84e-27 | -3.53e-29 | -4.78e-32 | 6.53e-33 | 1.04e-34 | 7.70e-37 | 2.94e-39 | 4.68e-42 |
| $k_6$ | 3.04e-09 | 2.36e-08 | 2.71e-10 | -1.92e-12 | -5.26e-13 | -3.02e-15 | 8.45e-16 | 2.01e-17 | -6.24e-20 | -3.64e-20 | -1.03e-22 | 2.54e-23 | 4.45e-25 | -4.80e-27 | -2.18e-28 | -1.70e-30 | 2.13e-32 | 5.39e-34 | 4.68e-36 | 1.90e-38 | 3.34e-41 |
| $k_7$ | 4.91e-07 | 1.11e-06 | 1.73e-08 | 2.26e-10 | 1.32e-11 | -1.42e-12 | -1.03e-13 | 4.25e-16 | 1.76e-16 | 3.01e-18 | -1.02e-19 | -3.84e-21 | -8.30e-24 | 1.50e-24 | 2.62e-26 | 8.39e-30 | -5.23e-30 | -7.93e-32 | -5.75e-34 | -2.16e-36 | -3.39e-39 |
| $k_8$ | 6.16e-05 | 6.08e-05 | 8.59e-07 | 1.73e-08 | -1.60e-12 | -1.03e-12 | -1.03e-12 | -6.72e-15 | 1.57e-15 | 3.95e-17 | -7.11e-19 | -4.09e-20 | -2.74e-22 | 1.37e-23 | 3.04e-25 | 8.62e-28 | -5.04e-29 | -8.66e-31 | -6.64e-33 | -2.59e-35 | -4.19e-38 |
| $k_9$ | 3.26e-03 | 1.25e-03 | 8.04e-05 | -2.95e-07 | -2.11e-08 | 2.97e-10 | 2.80e-11 | -1.18e-13 | -3.82e-14 | -5.50e-16 | 2.24e-17 | 7.41e-19 | 3.53e-22 | -3.02e-22 | -4.84e-24 | 3.64e-27 | 1.03e-27 | 1.49e-29 | 1.05e-31 | 3.88e-34 | 5.99e-37 |
| $k_{10}$ | 4.42e+01 | 5.50e-01 | 7.42e-03 | 5.02e-05 | -1.27e-08 | -1.29e-08 | 4.33e-10 | 5.82e-13 | -2.61e-15 | -5.97e-15 | 1.02e-16 | 4.88e-18 | 2.19e-20 | -1.62e-21 | -3.13e-23 | -4.75e-26 | 5.56e-27 | 8.82e-29 | 6.50e-31 | 2.47e-33 | 3.90e-36 |



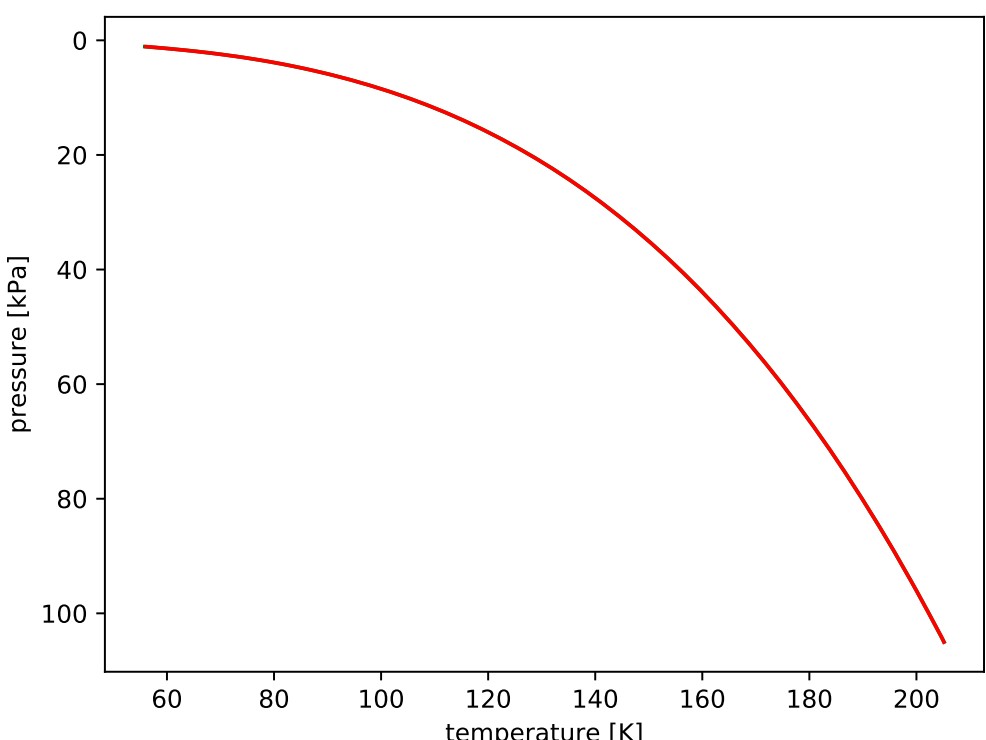

**Figure 2.** True (green) and modelled (red) $\theta_{ref}$ (difference between curves not apparent at this scale), for $\theta_{ref}$ equal to -70°C wet-bulb potential temperature.




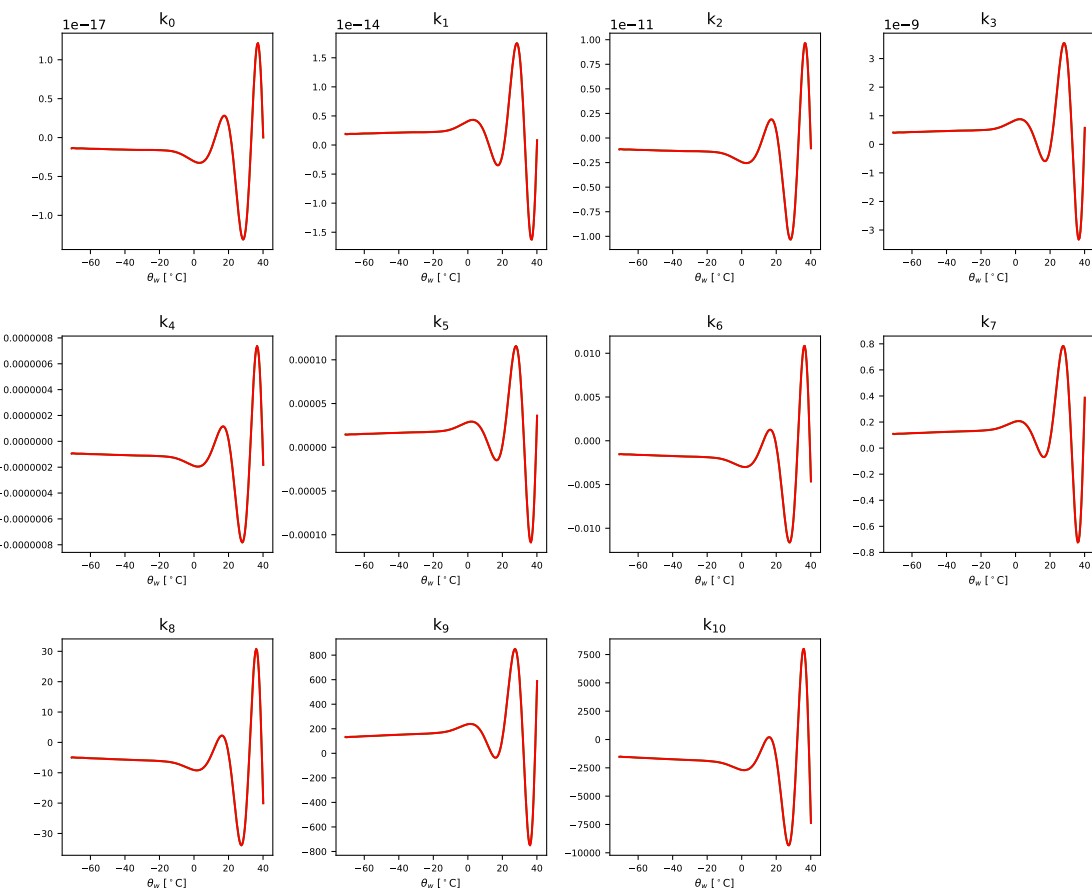

**Figure 3.** True and modelled polynomial fitting parameters (difference between curves not apparent at this scale). The ordinate is the dimensionless coefficient indicated above each graph.



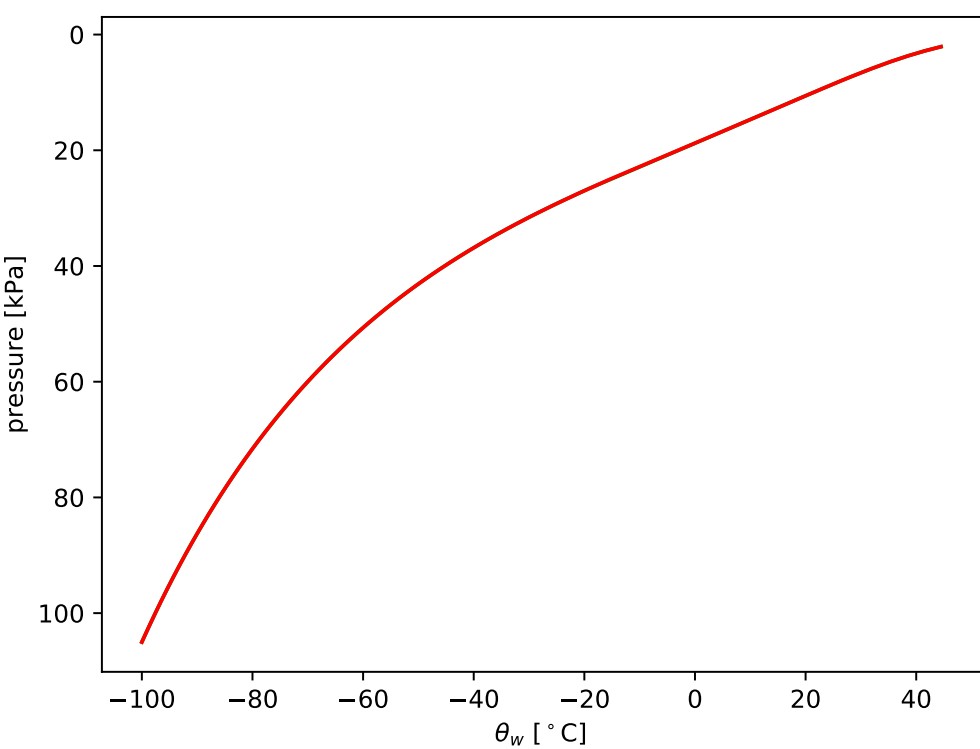

**Figure 4.** True and modelled $T_{ref}$ (difference between curves not apparent at this scale), for $T_{ref}$ equal to -100°C isotherm.




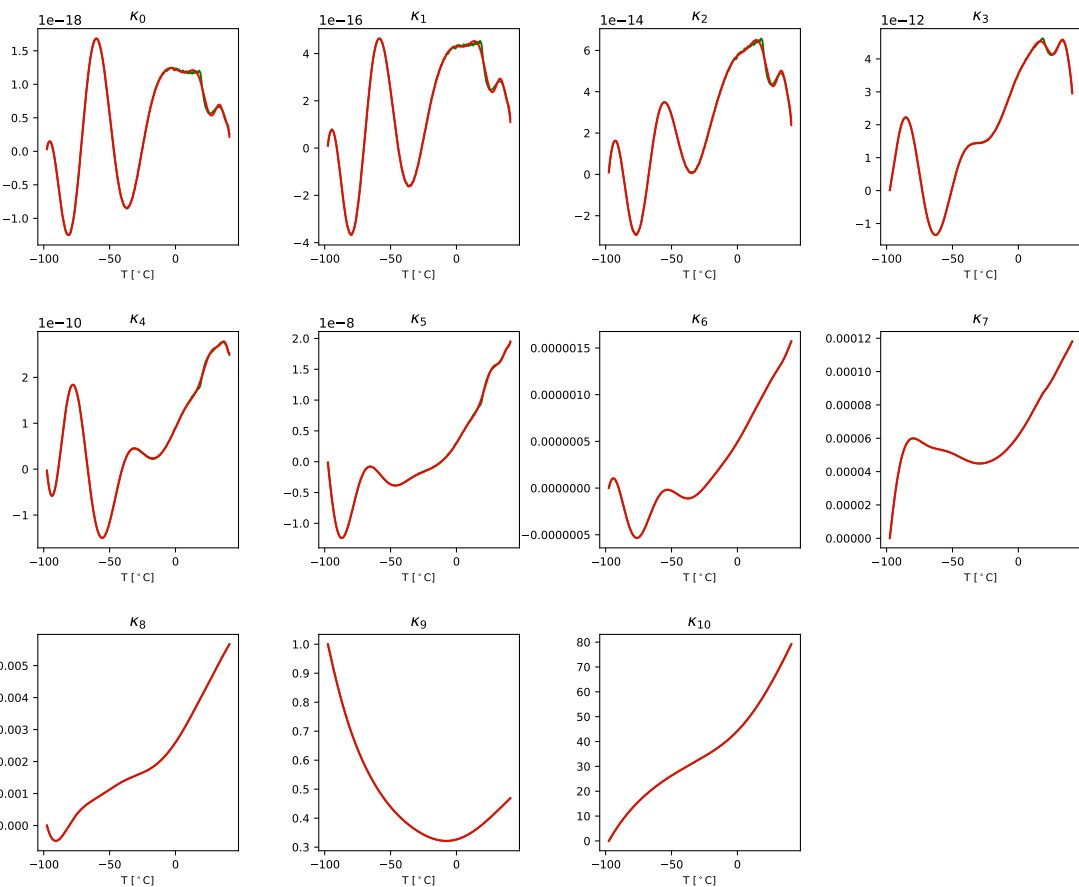

**Figure 5.** True (green) and modelled (red) polynomial fitting parameters. The abscissa is air temperature $T$. The ordinate is dimensionless.




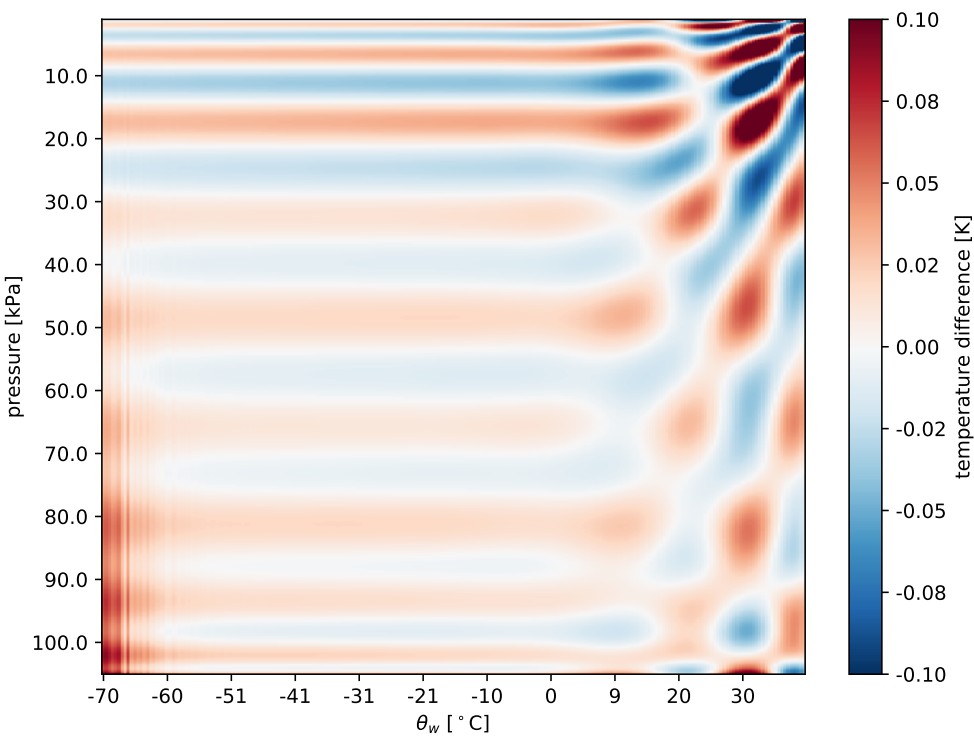

**Figure 6.** Approximation error between directly calculated and modelled $T$ along moist adiabats $\theta_w$.



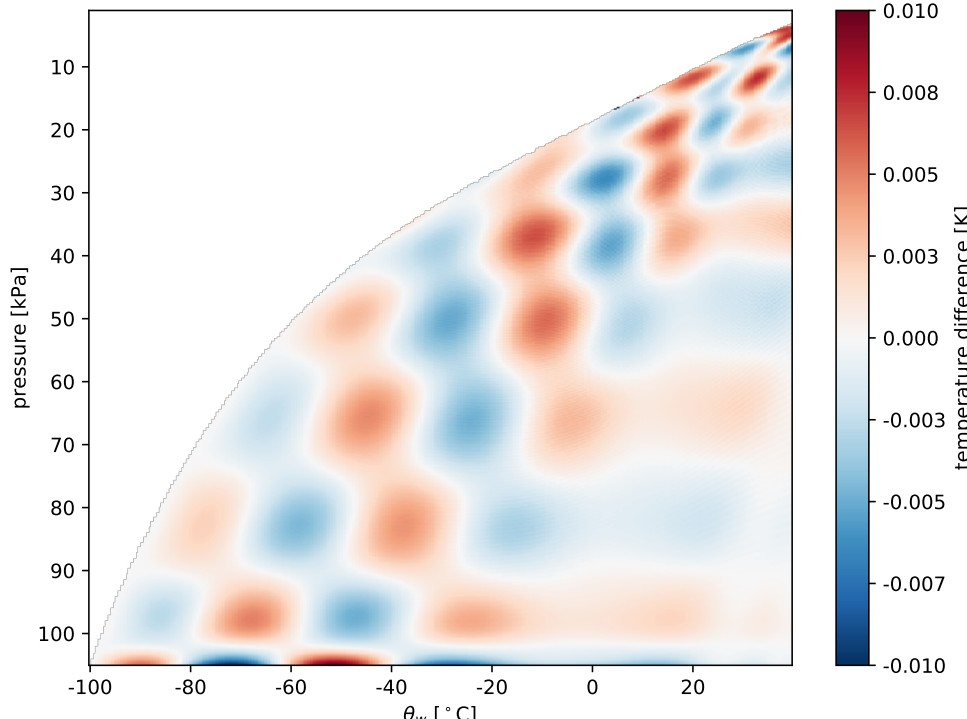

**Figure 7.** Approximation error between directly calculated and modelled $\theta_w$ along isotherms $T$.

*Competing interests.* No competing interests are present.

*Acknowledgements.* This work was supported by funding from Natural Sciences and Engineering Research Council (NSERC) PGS-D to N. Moisseeva, Discovery grants to R.Stull, with additional support from BC Hydro and Mitacs.



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
