# Peer review of "A noniterative approach to modelling moist thermodynamics"

_Atmospheric Chemistry and Physics, 2017_

## Referee Comment (RC1) · Anonymous Referee #1 · 9 Oct 2017

Review of 'A non-iterative approach to modelling moist thermodynamics' by Nadya Moisseeva and Roland Stull

The article presents an interpolation function to compute wet-bulb potential temperature as a function of pressure and temperature and its inverse, i.e. temperature as a function of pressure and wet-bulb potential temperature. The direct computation of wet-bulb potential temperature involves solving a nonlinear differential equation and therefore it can only be solved by iterative numerical methods. From this point of view, having a ready-made interpolant can be a valuable contribution within the scope of ACP for researchers and modellers. However, the manuscript can be improved by adding references and being more specific in the language used to present the mathematical description of the method. Perhaps more important is the definition of a set of 'true'

adiabats that serve as reference to develop the interpolant. How accurate is this 'truth' itself? I think this deserves further discussion. I elaborate on these recommendations in the comments below.

Specific comments

Table 1: How sensitive are the 'truth' adiabats to the values of the constants in Table 1? Also, what is the source of the constants' values? They do not always correspond to the values in e.g. Bolton (1980). Please, provide references.

P1L20: Include reference to Bolton (1980)

P2L21-L22: I don't see how normalizing the curves removes any inherent nonlinearity at all. Please explain further. Also give mathematical expressions for the operations you are doing here. I'm interpreting your division by a reference moist adiabat as $T(P, \theta_w)/T_{ref}(P)$, where $T_{ref}(P) = T(P, \theta_{ref})$ and $\theta_{ref} = -70°$C is a constant. Is this interpretation correct?

P2L24: What does 'the resulting transformed adiabats shift around the $\theta_{ref}$ unity line' mean? $\theta_{ref}$ is not unity and is not even close to it.

P2L27: In what sense are polynomials are well-behaved and why is this behaviour convenient?

P2L29: Chaotic behaviour is a property of dynamical systems and polynomials per se are not dynamical systems. So can you clarify what chaotic behaviour of high-degree polynomials are you referring to here? Please include references.

P3L7: Discuss further the results that you get with bi-exponential and arctan to explain what the accuracy is insufficient. How different is this accuracy to that achieved by your chosen method?

Section 4: The notation in very confusing. For instance, step 2 of Section 4.1, in which the computation of $T(P, \theta_w)$ is described, requires the computation of $\theta_{ref}(P)$.

However, $\theta_{ref}$ was assumed constant in Section 2.2!! I believe what you actually need to compute is $T_{ref}(P) = T(P, \theta_{ref})$, where $\theta_{ref}$ is a constant. A similar notation problem is present in Section 4.2.

Technical corrections

Title: It should read 'noniterative'

P1L20: It should read to 'To improve'

P2L17: It should say $\theta_w(P, T)$.

---

## Referee Comment (RC2) · Anonymous Referee #2 · 25 Oct 2017

This paper presents a technique for calculating temperature or wet bulb potential temperature along moist adiabats. Based on a high-order polynomial fit, the technique is considerably more accurate and less computationally burdensome than the iterative or look-up table procedures that are typically employed in most numerical weather prediction models. The high practical value of this work merits publication in ACP. The paper is well written and the methodology is clearly presented. I've provided a few minor suggestions for improvement below.

Main comments:

1) There are more figures (seven in total) than necessary for a short technical note. Figures 2-5 provide no information beyond the demonstration that the polynomial fits are indistinguishable from the "truth". The authors could consider removing these figures.

2) Some potentially useful context to add to the manuscript would be to address the question of whether errors associated with pre-existing methods are systematic or just noisy. Systematic errors in temperature would result in biased latent heating profiles, which could in turn have dynamical implications on the grid scale. If this were the case, then the improvements offered by the authors' methodology would be more substantial than a simple low-cost noise correction.

Other minor issues:

P1.L20: Spelling error. "improve" P3.L7-8. I don't understand why alternative function fits are "unlikely to be sufficiently accurate to be useful." Why not? What precisely does "well behaved parameters" mean here. The language used in this and the following paragraph is imprecise and the claims sound subjective. P4.L21-22. How much accuracy is compromised if the Table values are used instead of the spreadsheet. Can you put some numbers to this claim. Is the Table even necessary if the authors are cautioning against the implementation of the numbers in the Table? P6.L7-8 and L28-29 and elsewhere there are paragraphs comprised of single sentences. Can these sentences be merged with either the preceding or following paragraphs?

---

## Author Comment (AC1) · 9 Nov 2017

We thank the Referees for their constructive comments. Please find the original Referee notes in bold, our responses in italics and modifications to the manuscript in blue, below.

**Author Response to Referee #1**

The article presents an interpolation function to compute wet-bulb potential temperature as a function of pressure and temperature and its inverse, i.e. temperature as a function of pressure and wet-bulb potential temperature. The direct computation of wet-bulb potential temperature involves solving a nonlinear differential equation and therefore it can only be solved by iterative numerical methods. From this point of view, having a ready-made interpolant can be a valuable contribution within the scope of ACP for researchers and modellers. However, the manuscript can be improved by adding references and being more specific in the language used to present the mathematical description of the method. Perhaps more important is the definition of a set of 'true' adiabats that serve as reference to develop the interpolant. How accurate is this 'truth' itself? I think this deserves further discussion. I elaborate on these recommendations in the comments below.

Our responses below address the weaknesses of this technical note identified by the Referee.

**Specific comments**

Table 1: How sensitive are the 'truth' adiabats to the values of the constants in Table 1? Also, what is the source of the constants' values? They do not always correspond to the values in e.g. Bolton (1980). Please, provide references.

For our proposed solution, we tried to use more recent estimates of constant values than those suggested in Bolton (1980), where available. However, to provide a quantitative response to the above concern, we have repeated our iterative calculation of the "truth" adiabats using the values in Bolton (1980). The MAE between the two sets of solutions is 0.004°C. The figure below shows the distribution of error throughout the modelled domain.

Notably, the accuracy of the method we propose in this technical note is largely unaffected by the choice of "truth". Any change merely results in an adjustment of the polynomial fitting coefficients. Should more precise constant values and/or thermodynamic relationships become established in the future, our method can be re-applied to produce updated parameters without loss of accuracy (within the limits of the specific

**polynomial optimizer used). We have added the following comment in the Evaluation section of the manuscript to address this:**

The specific choice of thermodynamic constants and relationships undoubtedly effects the definition of "truth" used in this work, however, has little effect on the overall validity of the approach. Should more precise constant values and/or thermodynamic relationships become established in the future, the proposed method can be re-applied to generate updated fitting coefficients without loss of accuracy (within the limits of the specific polynomial optimization routine used).

**More textual changes:**

- Table 1 is updated with references for the two constants that differed from Bolton (1980)
- We adjusted the value of  $C_p$  to include more significant figures to agree with Bolton (1980)
- Reference temperature typo was corrected
- All figures as well as coefficient values in the Supplement were regenerated to reflect the additional significant figures in  $C_p$  value. This resulted in alternative rounding for MAE values for  $T(P,\theta w)$ , which changed from 0.017°C to 0.016°C. We've adjusted this value throughout the manuscript.
- The following note was added in-text, at the beginning of Section 2.1:

The constants used to devise this solution are consistent with Bolton (1980), unless otherwise indicated in Table 1.

**P1L20: Include reference to Bolton (1980)**

Added.

P2L21-L22: I don't see how normalizing the curves removes any inherent nonlinearity at all. Please explain further. Also give mathematical expressions for the operations you are doing here. I'm interpreting your division by a reference moist adiabat as  $T(P,\theta w)/Tref(P)$ , where  $Tref(P) = T(P,\theta ref)$  and  $\theta ref = -70$ °C is a constant. Is this interpretation correct?

We would first like to address the particular notation, to which the Referee points here and in later comments as well. Perhaps the source of confusion is the common meteorological convention, by which wetbulb potential temperature at standard pressure is used to label moist adiabats (see Ambaum 2010, p119; Stull 2017, p105 and p152-158). Such practice simplifies the use of thermodynamic diagrams, and is considered standard in applied meteorology. We, however, agree that some readers may be unfamiliar with such notation and it should be addressed explicitly. We've added the following brief statement in the beginning of Section 2.1:

Note, that throughout this technical note we will rely on a common meteorological convention, by which wet-bulb potential temperature at standard pressure is used to label moist adiabats. Such references, hence, represent curves, rather than constants and are bolded for clarity.

All references that rely on this notation in the manuscript have been changed to bold font to help differentiate them from constants.

**To address the remaining part of the comment:**

The mathematical interpretation proposed by the Referee is correct - the operation is a simple division  $T(P,\theta_w)/T(P,\theta_{ref})$ . This, by the meteorological convection noted above, corresponds to  $\theta_w/\theta_{ref}$ , described in the first version of the manuscript on P2L22: "...we can normalize our curves by dividing each  $\theta_w$  by a reference moist adiabats  $\theta_{ref}$ ". In practice, however, this normalization is equivalent to fitting  $\theta_w$  as a function of  $\theta_{ref}$ . Overall, this is a purely technical trick employed to help the optimizer converge. As we are trying to capture all the moist adiabats with a single function, it helps to have the curves behave as similar as possible. Yet the moist adiabats represent a non-linear process, so there's no simple way (we tried!) to collapse them into a single shape, i.e. achieve some sort of similarity theory. Instead we fit one of the solutions ( $\theta_{ref}$ ), and then model the deviations from that fit.

**We've changed the beginning of Section 2.2 to describe this operation more precisely:**

While the moist adiabatic curves  $\theta_w$  in Figure 1 look smooth and fairly similar, it is challenging for most common optimization routines to capture all of them with one analytical expression with high accuracy. Due to the inherently non-linear nature of the process, there is no simple way to collapse the curves into a single shape. However, to aid fitting, we can normalize our curves by modelling  $\theta_w$  as a function of a reference moist adiabat  $\theta_{ref}$ . That allows us to model only the deviations from a reference curve. For our example we used  $\theta_{ref} = -70^{\circ}$ C. This particular choice of  $\theta_{ref}$  implies no theoretical importance...

**P2L24: What does 'the resulting transformed adiabats shift around the θref unity line' mean? θref is not unity and is not even close to it.**

This is fairly simple to visualize by comparing plots of transformed adiabats produced with  $\theta_{ref} = -70^{\circ}$ C vs.  $\theta_{ref} = +40^{\circ}$ C. The blue and red correspond to positive and negative values, respectively. Select adiabats are shown for clarity. The **transformed** curve corresponding to  $\theta_{ref}$  vs.  $\theta_{ref}$  is indeed a unity line.

P2L27: In what sense are polynomials are well-behaved and why is this behaviour convenient?

**We have added the following clarification to the manuscript to highlight the benefits of using polynomial functions:**

We use polynomial fitting to describe T(P) for the fixed  $\theta_{ref}$ . This is convenient, since polynomials are generally well-behaved and are computationally easy to use. In particular, they are both continuous and smooth, while being able to capture a wide variety of shapes. Moreover, they have well understood properties and a simple form, allowing the model to be easily implemented in a basic spreadsheet.

Additional discussion on the particular properties of polynomial fitting important to our problem is provided as response to a later Referee comment on use of bi-exponential and arctan alternatives.

**P2L29: Chaotic behaviour is a property of dynamical systems and polynomials per se are not dynamical systems. So can you clarify what chaotic behaviour of high-degree polynomials are you referring to here? Please include references.**

The term "chaotic behavior" indeed carries a much more specific mathematical meaning than we intended to imply. A more accurate description of the particular property of high-degree polynomials we wanted to address is their tendency to oscillate wildly. We updated the manuscript to remove the inaccurate use of the term and provide a brief clarification as follows:

...Since we are examining a fixed range of temperatures relevant to atmospheric applications, the potentially extreme oscillatory behavior of high-degree polynomials outside of the modelled domain is not a primary

concern. The fitted polynomials have no predictive value outside of the modelled range and serve purely as an interpolation function. While the large number of possible inflection points associated with high degree polynomials may be of a concern near the edges of the fitting interval, a problem known as Runge's phenomenon (Epperson, 1987), the current algorithm relies on least-squares method to minimize the effect and achieve a high quality fit. For this example, the aim was to ensure that the mean absolute error (MAE) is on the order of  $10^{-2}$  degrees C...

**P3L7: Discuss further the results that you get with bi-exponential and arctan to explain what the accuracy is insufficient. How different is this accuracy to that achieved by your chosen method?**

Thank you for pointing out the lacking discussion of other fitting functions. In part, the evasive nature of our comment on alternative approaches is due to the difficultly in providing a true quantitative comparison of the different methods. While we were able to model the transformed curves with bi-exponential and arctan functions and achieve a visually-satisfactory fit with errors on the order of degrees, these fits do not constitute a complete solution. The remaining part of the problem is modelling the variable parameters kn. This is where the well-behaved nature of polynomials mentioned earlier plays a key role. Unlike polynomials, bi-exponential and arctan functions produce parameter curves, which are neither smooth, nor continuous. Most are asymptomatic, rendering them near-impossible to model with polynomials. Yet another set of alternative functions would be necessary to capture the parameter curves, which, combined with the errors already present from fitting the transformed adiabats would produce a complex and discontinuous error field. In essence, these alternative attempts are equivalent to trying to manually perform GEP, which was already [properly] done by Bakhshaii and Stull (2013). Their method resulted in errors, which were both discontinuous and larger than with our proposed approach.

Hence, while we are unable to provide a quantitative comparison of the different approaches to fitting, we hope that the following changes to the manuscripts address this point in a more clear and complete manner: For this work we tested bi-exponential, arctan, rational and polynomial functions. Generally, a reasonable (on the order of 1-2°C) fit can be achieved with both bi-exponential and arctan functions using as little as three variable parameters. While efficient, the results of such fit are unlikely to be sufficiently accurate to be useful for real-life modelling applications and, more importantly, only constitute a part of the solution. The bigger concern with these choices is that, unlike polynomials, they produce variable parameters that do not appear well-behaved. Discontinuity and asymptomatic behavior arising from error minimization for all transformed adiabats render the parameter curves very difficult to model. A variety of functions would be necessary to capture the parameter behavior, which in turn is likely to produce a complex and discontinuous error field, such as appeared in Bakhshaii and Stull (2013).

**Section 4: The notation in very confusing. For instance, step 2 of Section 4.1, in which the computation of $T(P,\theta w)$ is described, requires the computation of $\theta ref(P)$ .**

However,  $\theta$  ref was assumed constant in Section 2.2!! I believe what you actually need to compute is Tref (P) = T (P,  $\theta$  ref ), where  $\theta$  ref is a constant. A similar notation problem is present in Section 4.2.

We hope that our earlier response on the use of meteorological notation has addressed the Referee's concern.

**Technical corrections Title: It should read 'noniterative'**

Corrected, with much self-reproach.

**P1L20: It should read to 'To improve'**

Corrected.

**P2L17: It should say θw (P, T).**

Corrected.

REFERENCES

Ambaum, M.: Thermal Physics of the Atmosphere, John Wiley, New York, 2010.

Bakhshaii, A. and Stull, R.: Saturated pseudoadiabats-A noniterative approximation, Journal of Applied Meteorology and Climatology, 52, 5–15, doi:10.1175/JAMC-D-12-062.1, 2013.

Bolton, D.: The Computation of Equivalent Potential Temperature, Monthly Weather Review, 108, 1046–1053, doi:10.1175/1520-0493(1980)108<1046:TCOEPT>2.0.CO;2, https://doi.org/10.1175/1520-0493(1980)108<1046:TCOEPT>2.0.CO;2, 1980.

Burns, L.: Range Reference Atmosphere 2013 Production Methodology, Tech. rep., NASA, 2015.

Epperson, J. F.: On the Runge Example, Am. Math. Monthly, 94, 329–341, doi:10.2307/2323093, http://dx.doi.org/10.2307/2323093, 1987.

Stull, R.: Practical Meteorology: An Algebra-based Survey of Atmospheric Science, ISBN 978-0-88865-283-6, Free online for everyone worldwide, under a Creative Commons license, https://www.eoas.ubc.ca/books/Practical\_Meteorology/, 2017.

**Author Response to Referee # 2**

This paper presents a technique for calculating temperature or wet bulb potential temperature along moist adiabats. Based on a high-order polynomial fit, the technique is considerably more accurate and less computationally burdensome than the iterative or look-up table procedures that are typically employed in most numerical weather prediction models. The high practical value of this work merits publication in ACP. The paper is well written and the methodology is clearly presented. I've provided a few minor suggestions for improvement below.

We thank the Referee's acknowledgement of the practical value of this note. We hope our responses below address the manuscript's shortcomings.

**Main comments:**

1) There are more figures (seven in total) than necessary for a short technical note. Figures 2-5 provide no information beyond the demonstration that the polynomial fits are indistinguishable from the "truth". The authors could consider removing these figures.

We are happy to eliminate Figures 2-5, as per Referee's suggestion.

2) Some potentially useful context to add to the manuscript would be to address the question of whether errors associated with pre-existing methods are systematic or just noisy. Systematic errors in temperature would result in biased latent heating profiles, which could in turn have dynamical implications on the grid scale. If this were the case, then the improvements offered by the authors' methodology would be more substantial than a simple low-cost noise correction.

In our view, existing methods are likely to contain both random and systematic errors. We have added the following discussion to the manuscript to provide more detail:

While interpolating values from look-up tables generally results in random errors, iterative solutions with a coarse step could potentially suffer from a directional drift due to numerical integration errors, which may introduce a consistent bias into latent heating profiles. Moreover, near the top of the atmosphere, where each pressure step corresponds to a large temperature jump along the moist adiabats the numerical solutions tend to become unstable. Though both of these concerns are addressed with the proposed low-cost polynomial method, the broader challenge of our limited overall understanding of moist convection remains. Existing thermodynamic relationships are based on the assumption of either a reversible moist adiabatic or an irreversible pseudoadiabatic process. Real world atmospheric processes are likely to be a combination of both (Iribarne and Godson, 1981). The uncertainty introduced by our limited knowledge of the true state of saturated air is likely to remain the central obstacle in capturing moist convection.

**Other minor issues: P1.L20: Spelling error. "improve"**

Corrected.

**P3.L7-8.** I don't understand why alternative function fits are "unlikely to be sufficiently accurate to be useful." Why not? What precisely does "well behaved parameters" mean here. The language used in this and the following paragraph is imprecise and the claims sound subjective.

We agree with the Referee that our approach to describing the alternative fitting functions is very vague. This point was also brought up by Referee #1. While we are unable to provide a precise quantitative answer, we hope that our response (copied below) will address this point in a clearer manner: Thank you for pointing out the lacking discussion of other fitting functions. In part, the evasive nature of our comment on alternative approaches is due to the difficultly in providing a true quantitative comparison of the different methods. While we were able to model the transformed curves with bi-exponential and arctan functions and achieve a visually-satisfactory fit with errors on the order of degrees, these fits do not constitute a complete solution. The remaining part of the problem is modelling the variable parameters kn. This is where the well-behaved nature of polynomials mentioned earlier plays a key role. Unlike polynomials, bi-exponential and arctan functions produce parameter curves, which are neither smooth, nor continuous. Most are asymptomatic, rendering them near-impossible to model with polynomials. Yet another set of alternative functions would be necessary to capture the parameter curves, which, combined with the errors already present from fitting the transformed adiabats would produce a complex and discontinuous error field. In essence, these alternative attempts are equivalent to trying to manually perform GEP, which was already [properly] done by Bakhshaii and Stull (2013). Their method resulted in errors, which were both discontinuous and larger than with our proposed approach.

Hence, while we are unable to provide a quantitative comparison of the different approaches to fitting, we hope that the following changes to the manuscripts address this point in a more clear and complete manner: For this work we tested bi-exponential, arctan, rational and polynomial functions. Generally, a reasonable (on the order of 1-2°C) fit can be achieved with both bi-exponential and arctan functions using as little as three variable parameters. While efficient, the results of such fit are unlikely to be sufficiently accurate to be useful for real-life modelling applications and, more importantly, only constitute a part of the solution. The bigger concern with these choices is that, unlike polynomials, they produce variable parameters that do not appear well-behaved. Discontinuity and asymptomatic behavior arising from error minimization for all transformed adiabats render the parameter curves very difficult to model. A variety of functions would be necessary to capture the parameter behavior, which in turn is likely to produce a complex and discontinuous error field, such as appeared in Bakhshaii and Stull (2013).

**P4.L21-22. How much accuracy is compromised if the Table values are used instead of the spreadsheet. Can you put some numbers to this claim. Is the Table even necessary if the authors are cautioning against the implementation of the numbers in the Table?**

We were hesitant to provide the coefficient tables in-text form the start, and chose to include them for completeness only. We agree with the Referee that they are not particularly valuable, as most readers will use the spreadsheet for both: high accuracy and ease of data formatting. We, therefore, removed the tables from the manuscript.

**P6.L7-8 and L28-29 and elsewhere there are paragraphs comprised of single sentences. Can these sentences be merged with either the preceding or following paragraphs?**

Line breaks were removed.

**REFERENCES**

Bakhshaii, A. and Stull, R.: Saturated pseudoadiabats-A noniterative approximation, Journal of Applied Meteorology and Climatology, 52, 5–15, doi:10.1175/JAMC-D-12-062.1, 2013.

Iribarne, J. and Godson, W., eds.: Atmospheric Thermodynamics, chap. IX, pp. 176–244, Springer Netherlands, 2 edn., doi:10.1007/978-94-009-8509-4, 1981.

---

## Author Comment (AC2) · 9 Nov 2017

**A noniteratiave noniterative approach to modelling moist thermodynamics**

**Nadya Moisseeva1 and Roland Stull1**

[revised manuscript text omitted]

**7 Figures**

**Figures**